# Transcatheter Device Therapy and the Integration of Advanced Imaging in Congenital Heart Disease

**DOI:** 10.3390/children9040497

**Published:** 2022-04-02

**Authors:** Abhay A. Divekar, Yousef M. Arar, Stephen Clark, Animesh Tandon, Thomas M. Zellers, Surendranath R. Veeram Reddy

**Affiliations:** 1Department of Pediatrics, University of Texas Southwestern Medical Center, 5323 Harry Hines Blvd., Dallas, TX 75390, USA; yousef.arar@utsouthwestern.edu (Y.M.A.); stephen.clark@utsouthwestern.edu (S.C.); thomas.zellers@utsouthwestern.edu (T.M.Z.); suren.reddy@utsouthwestern.edu (S.R.V.R.); 2Pediatric Cardiology, Children’s Medical Center, 1935 Medical District Drive, Dallas, TX 75235, USA; 3Department of Pediatric Cardiology, Cleveland Clinic Children’s Hospital, Cleveland, OH 44195, USA; tandona2@ccf.org

**Keywords:** congenital heart disease, transcatheter intervention, cross-sectional imaging, stents, transcatheter pulmonary valve implantation, Septal Occluders

## Abstract

Transcatheter device intervention is now offered as first line therapy for many congenital heart defects (CHD) which were traditionally treated with cardiac surgery. While off-label use of devices is common and appropriate, a growing number of devices are now specifically designed and approved for use in CHD. Advanced imaging is now an integral part of interventional procedures including pre-procedure planning, intra-procedural guidance, and post-procedure monitoring. There is robust societal and industrial support for research and development of CHD-specific devices, and the regulatory framework at the national and international level is patient friendly. It is against this backdrop that we review transcatheter implantable devices for CHD, the role and integration of advanced imaging, and explore the current regulatory framework for device approval.

## 1. Introduction

We have made tremendous strides in the diagnosis and treatment of congenital heart defects (CHD) [1]. There has been a paradigm shift in the comprehensive management of all CHDs. Cardiac catheterizations are now increasingly performed as therapeutic procedures, and in many cases a permanent implant is placed. These procedures are performed in a very heterogenous population ranging in size from small and fragile premature infants to adults with CHD. While there has been an increase in the availability of innovative CHD-specific devices, off-label use of approved devices remains an essential component in transcatheter treatment of CHD.

Advances in diverse fields including development of novel materials (smart metals), additive manufacturing, and understanding specific implantation site characteristics have contributed to the explosive growth in transcatheter delivery of permanent implants [2,3]. The dynamic cardiovascular milieu poses demanding challenges for imaging, as well as the ability of the implant to tolerate repetitive cyclical stress while still maintaining structural integrity [4]. Improved temporal and spatial resolution of imaging modalities allows for improved understanding of device implantation site interaction [5]. This information is used for computational modeling and simulation which in turn helps determine material properties and device design [6,7]. Advances in transthoracic echocardiography (TTE), transesophageal echocardiography (TEE), intracardiac echocardiography (ICE), and intravascular ultrasound (IVUS) allows for real-time three-dimensional (3D) imaging and tissue characterization [8]. With significant reduction in radiation dose, the utilization of cardiac computed tomography (CT) has increased. This allows critical insight into complex anatomy and relationships amongst adjacent structures. For example, high-quality pre-procedural CT imaging is critical for patient selection, device fit, and determining optimal vascular access site prior to transcatheter pulmonary valve implantation and ductus arteriosus stenting [6,9]. Cardiovascular magnetic resonance imaging (cMRI) provides excellent tissue characterization, volumetric assessment, 3D dynamic acquisitions, and 4D flow assessment without exposing patients to ionizing radiation. Cardiac CT and cMRI 3D datasets can be repurposed for generating 3D printed anatomic models (3DPAM) and virtual reality datasets to conceptualize and test creative interventional procedures [10]. Advanced imaging datasets can be integrated with fluoroscopy, allowing the implanter “to see” the anatomy and thereby improving precision during device deployment and ultimately patient outcomes [11].

High procedural volumes in other interventional sub-specialties such as adult interventional catheterization, interventional radiology, and neurosurgery have allowed for multiple device iterations and miniaturization. The congenital heart disease space has indirectly benefited by innovatively adapting (off-label use) these devices for use in children. For example, miniaturization of vascular introducers for transradial coronary interventions provides an introducer with the smallest outer diameter that can be used in neonates and small infants [12]. Finally, industry and regulatory agencies are now working closely with academia to optimize device development for pediatric-specific devices [13,14,15].

In this review the authors aim to:A.Provide an overview of current transcatheter implantable devices and briefly discuss newer devices being developed.B.Discuss the role and integration of advanced imaging in transcatheter interventions.C.Explore the current patient friendly regulatory framework encouraging innovative device development.

This review is not intended to provide an exhaustive analysis of each device.

## 2. Overview of Current Transcatheter Therapies for CHD

The explosive growth of cardiovascular implants has caused a paradigm shift in the approach to management of CHD. Defects which once required surgical repair are now routinely treated with transcatheter intervention. Nowhere is this better demonstrated than in transcatheter closure of secundum atrial septal defects and patent ductus arteriosus. In many instances, transcatheter therapies work in conjunction with surgery to optimize patient outcomes. This is best seen with the explosion of endovascular stenting and transcatheter valve therapies. With this backdrop we will review currently used permanent implants for CHDs. For convenience, we will group these therapies into four sections.

Device closure of cardiac shuntsEndovascular stenting for obstructed blood vesselsTranscatheter valve therapiesVascular occlusion devices

### 2.1. Device Closure of Cardiac Shunts

Traditionally, surgical closure for atrial and ventricular septal defects (ASDs and VSDs) and patent ductus arteriosus (PDA) was the standard approach. Now, transcatheter closure is routine and considered as first line therapy for closure of most secundum ASDs, PDAs, and many muscular VSDs.

#### 2.1.1. Atrial Septal Defect Closure

ASD is one of the most common CHD, both in isolation (6–10/10,000 live births) and in association with complex CHD [16]. Left to right atrial shunting of blood through moderate to large defects causes right atrial and ventricular enlargement and pulmonary over-circulation. This volume overload, while often asymptomatic in early childhood, can cause poor growth, frequent pulmonary infections, and exercise limitations in some patients. In later life, untreated significant ASDs can cause atrial and ventricular arrhythmias, pulmonary hypertension, heart failure, and increased mortality [16]. Therefore, ASD closure is universally recommended during childhood [17,18].

In 1976 King and Mills introduced the double umbrella device for ASD closure [19]. The first commercially available ASD Occluder in the US was the CardioSEAL Occlusion Device (Nitinol Medical Technologies, Inc., Boston, MA, US, no longer manufactured), followed by the Amplatzer Septal Occluder, the Gore Helex Septal Occluder (no longer manufactured), and finally the Gore^R^ Cardioform family of devices.

In the United States, there are currently two FDA approved transcatheter ASD closure devices—the Amplatzer Septal Occluder (ASO; Abbott, Abbott Park, IL, USA) and the Gore Cardioform ASD Occluder (GCSO; WL Gore, Flagstaff, AZ, USA) (Figure 1 and Figure 2). Both devices have a nitinol frame. Closure is accomplished by a central waist, with connecting left and right atrial discs holding the device in position. Both devices can be retrieved and repositioned before final release. Adequate septal rims (>5 mm) are recommended for device closure. While the retro-aortic rim is frequently deficient, successful ASD closure is usually possible in a vast majority of patients. On the other hand, a deficient inferior or postero-inferior rim significantly increases the risk of device embolization, especially in large defects and/or when associated with other rim deficiencies [20]. Both devices approved for ASD closure in the United States have demonstrated excellent safety and efficacy [21].

Superior sinus venosus (SV) ASDs with partial anomalous pulmonary venous return are surgically repaired. In 2013, Abdullah et al. first reported successful transcatheter closure using a covered stent [22]. The approach involves placement of a large diameter covered stent of appropriate length. The stent is differentially expanded; superiorly it is anchored in the superior vena cava and in the upper atrium it is expanded further to close the SVASD (funnel shaped appearance of the stent with the broad end in the atrium). The anomalous pulmonary veins drain around the stent into the left atrium. Careful assessment is performed prior to and after stent placement to make sure that the anomalous pulmonary venous drainage is unobstructed. Since this initial description, several operators have performed successful transcatheter closure of SVASD in adults, and the technique continues to mature. An international registry has been established to share best practices among implanters and provide long-term outcome data for this novel approach [23].

Complications and Knowledge Gaps:i.Erosion: The Amplatzer Septal Occluder has been associated with cardiac erosion causing pericardial effusion with or without cardiac tamponade, and rarely death [24,25]. While most cases of cardiac erosion occur early, erosions occurring several years after device closure have been reported [26]. Until 2020, the Gore family of ASD devices were not associated cardiac erosion; now two cases have been reported [27]. While a number of risk factors for cardiac erosion have been proposed, none are predictive of erosion [25,28,29]. ASD closure devices used outside the United States have also been associated with cardiac erosion [30,31].ii.Atrioventricular (AV) block: ASD Occluders have been associated with AV block [32]. Bink et al. have reported 15% incidence of preoperative first degree AV block; therefore careful preoperative evaluation with an electrocardiogram is important [33]. ASDs are also associated with risk of progressive AV block if genetically predisposed [34]. From an interventional standpoint, interaction between the device and the AV node in the triangle of Koch either from direct trauma or in response to inflammatory reaction have been postulated as possible mechanisms [32,35,36]. Proposed risk factors include large device (based on patient age, absolute device size, device to septal length ratio, device to height ratio), deficient postero-inferior rim, and direct contact of the device with the triangle of Koch following deployment [35]. Optimal approach for treatment of post-device closure AV block is controversial. Options include device retrieval and placement of a smaller device, medical therapy with high dose aspirin and steroids, and immediate device removal with surgical closure [32,35,36]. Treatment is likely best individualized, with medical therapy reserved for lower grades of AV block, and removal indicated for progressive AV block, failed medical therapy, or complete heart block.iii.For transcatheter SVASD closure longer-term data will be necessary to determine optimal stent design, how to avoid residual shunting, optimal closure strategy for residual defects, and monitoring for development of late pulmonary vein obstruction or sinus node dysfunction.

While there are only two approved devices in the United States, there are several approved ASD Occluders available in Europe and Asia. They include the Cocoon Septal Occluder (Vascular Innovations CO, Nonthaburi, Thailand), the Occlutech ASD Occluder (Occlutech, Schaffhausen, Switzerland), and the Lifetech CeraFlex ASD occluder (LifeTech Scientific Co., Shenzhen, China). All devices have been shown to be safe with good technical success and closure rates [37,38,39].

The Carag Bioresorbable Septal Occluder (Carag AG, Baar, Switzerland) is the first transcatheter septal occluder with bioresorbable metal-free framework. Two opposing polyester covers are attached to the framework. Once healed, the device framework is resorbed. A first-in-human trail conducted in Germany showed good results and the device has received CE-Mark [40]. The device is undergoing a prospective multicenter trial in the United States under Investigational Device Exemption Status (reSept Septal Occluder ASCENT-ASD, NCT04591392).

#### 2.1.2. Ventricular Septal Defect Closure

VSDs in isolation or as a component of complex heart disease are the most common CHD [41]. The frequency of transcatheter VSD closure is less frequent than closure of ASDs and PDAs. This is in part due to high frequency of spontaneous closure of muscular VSDs and the lack of an ideal perimembranous (pm) VSD closure device. In the United States, device closure of muscular (m) VSDs is generally accepted as an alternative to cardiac surgery [42,43]. On the other hand, the standard of care for pmVSDs in the United States remains surgical closure secondary to the unacceptably high rate of heart block with currently available devices.

Transcatheter mVSD closure in the United States is performed using either the Amplatzer mVSD Occluder or the off-label use of other Amplatzer devices. The approved Amplatzer mVSD device is a self-expandable double disc made of nitinol mesh available in sizes ranging from 4 to 18 mm (size of the waist) and delivered through a 5–9 Fr delivery sheath (Figure 3). Hemodynamically significant VSDs are more common in infants; these can be either closed percutaneously (transcatheter approach using an arteriovenous loop) or using a hybrid perventricular approach [42,44]. In small infants, the risk of hemodynamic instability and injury to cardiac valves is high, and therefore a hybrid perventricular approach is often chosen [45].

Complications and Knowledge Gaps:i.The reported incidence of complete heart block after pmVSD closure varies from 0.1–22% [46,47]. The devices associated with the highest risk are no longer used. For reference, the risk of complete heart block after surgical repair is less than 2% [47]. Therefore longer-term data is necessary before pmVSD closure becomes standard of care.ii.Injury to the aortic valve and both AV valves (especially tricuspid) can be seen especially after transcatheter pmVSD closure. Again, longer-term data is lacking, and this information is necessary for acceptance of routine pmVSD closure.iii.Residual shunting is reported in 3–29% of cases; for certain devices such as the Nit-Occlud Lê VSD-Coil (PFM Medical AG, Cologne, Germany) the incidence is even higher [47]. Longer-term follow-up will be necessary to determine if these residual shunts are hemodynamically significant and if intravascular hemolysis is a clinical problem.iv.During perventricular hybrid mVSD closure, there is risk of injury to the left ventricular wall causing perforation and bleeding and the risk of pseudoaneurysm formation [44,48].

While technical results for closing pmVSDs with specifically developed Amplatzer devices was good, there was an unacceptable incidence of heart block, and these devices are no longer available [49]. When the pmVSD is associated with development of an aneurysm or windsock in the membranous area, the Amplatzer family of devices (Amplatzer Ductal Occluder I, II and the Amplatzer Vascular Plug II) have been successfully used (off-label) for closure without causing heart block [50]. Studies from Asia and Europe are reporting experience with successful pmVSD closure using various devices including the Nit-Occlud Lê VSD-Coil (PFM Medical AG, Cologne, Germany), Lifetech Konar-MF (multi-functional) occluder (Lifetech, Shenzhen, China), CeraTM VSD Occluder (LifeTech Scientific, Shenzhen, China), and Occlutech PmVSD Occluder (Occlutech, Schaffhausen, Switzerland) [46,51,52].

#### 2.1.3. Patent Ductus Arteriosus Closure

In this section we will discuss transcatheter treatment of an isolated patent ductus arteriosus (PDA). PDA closure is indicated for hemodynamically significant left to right shunting which can manifest as asymptomatic left ventricular volume overload with or without clinical heart failure, failure to thrive, pulmonary hypertension, and significant lung disease in the premature infant. Transcatheter PDA closure was first described by Porstmann and colleagues in 1967 [53]. Over the last several decades, tremendous advances in device technology and delivery have allowed for safe and effective closure of PDA’s in a majority of patients; transcatheter closure is now the standard of care (except in premature infants) [54]. Originally requiring sheaths as large as 18 Fr, transcatheter PDA closure can now be performed in premature infants (>700 gm) using an FDA-approved Amplatzer Piccolo™ Occluder (Abbott, Minneapolis, MN, US) through a 4 Fr sheath [55,56].

Transcatheter closure of PDA is now first line therapy. The Amplatzer Duct Occluder I (ADO-I) was the first widely used (FDA approved 2003) purpose-built device for transcatheter closure of type A conical PDAs for patients weighing more than 6 kg. The ADO-I device is a mushroom-shaped self-expandable and repositionable device that is made of a nitinol wire mesh filled with polyester fabric to enhance thrombogenicity. For many PDA morphologies, especially tubular or elongated types, off-label use of other Amplatzer devices has allowed for successful and effective closure. These include the Amplatzer Vascular Plug II, Amplatzer Duct Occluder II (ADO II and ADO II AS), and the Amplatzer Muscular VSD and Septal Occluders for large PDAs. Free release and retrieval embolization coils have been extensively used for PDA closure (MReye Embolization Coils and MReye Flipper detachable embolization coil and delivery system, Cook Medical Bloomington, IN, US). The Nit-Occlud PDA (PFMMedical, Cologne, Germany) is another FDA approved nitinol coil designed for PDA closure [57].

Complications and Knowledge Gaps:i.Device embolization: The risk of device embolization was higher when coils were the only option; now with current technology the risk is lower. The ADO-I can be challenging to retrieve since the micro-screw is recessed. This is less problematic for the other Amplatzer devices.ii.Residual shunt with or without hemolysis is another complication that was more common when coils were routinely used. Fortunately, this risk is now very small.iii.Device related left pulmonary artery stenosis is an important consideration especially in small infants. Intraductal placement of the device and using a device without a disc on the pulmonary artery side are useful strategies.iv.Device-related aortic coarctation can occur. The risk is highest in small infants and neonates. Mild isthmic hypoplasia, protrusion of the device into the aorta, and short length of the PDA are some of the risk factors.v.Arterial injury: Until recently, arterial access was needed to perform assessment for residual shunting and arch obstruction prior to device release. The risk of arterial injury is highest in small infants. In premature infants, arterial access is not obtained; assessment for left pulmonary artery stenosis, aortic obstruction, and residual shunting is performed using transthoracic echocardiography. Many operators are now extending this experience to larger infants, further increasing the safety of the procedure.

Occlutech PDA occluder (Occlutech International AB, Helsingborg, Sweden) and Cera PDA Occluder (Lifetech Scientific Co. Ltd., Shenzhen, China) are some of the other devices available outside the US.

PDA closures in extremely small premature babies weighing more than 700 g is increasingly performed. The Amplatzer Piccolo™ Occluder is the only FDA approved device for premature infants. Good results have also been reported with the off-label use of the Medtronic Microvascular plug (Medtronic, Minneapolis, MN, US) and the Micro Plug set (KA Medical, Minneapolis, MN, USA) in premature infants (Figure 4) [58,59,60]. In premature infants, there is controversy regarding indications for PDA closure, optimal device, and surgical versus transcatheter closure options. Upcoming clinical trials are being designed to address these questions.

### 2.2. Endovascular Stenting of Obstructed Blood Vessels

Stent implantation/stent angioplasty (SI) is increasing used to treat hemodynamically significant vascular stenosis, to create intracardiac shunts, and as a framework for transcatheter valve implantation [61]. Even though SI is a commonly performed intervention in patients with CHD, there are very few FDA approved devices for use in CHD. Consequently, the interventional community has creatively adapted existing FDA approved devices. It is now generally agreed that SI is superior to balloon angioplasty (BA) both in terms of safety profile and acute success for treatment of vascular stenosis [62]. An important limitation of SI is the creation of “acquired stenosis” secondary to somatic growth and the inability to dilate small- and medium-diameter stents to necessary adult diameters.

Stents maintain luminal diameter by resisting vessel/tissue recoil. There are numerous stent designs, but the basic construct of all stents is the repeating collection of basic units called cells which are connected by hinges or joints. There are many classifications of stents including balloon expandable versus self-expanding, open cell versus closed cell, bare metal versus drug-eluting, small, medium, large, and extra-large diameter stents, covered versus uncovered, and pre-mounted versus hand-mounted stents. Each stent has unique characteristics related to the dilation potential, radial strength (ability to resist vessel recoil), flexibility or deliverability, and crossing profile (size of the delivery sheath).

Stent implantation is now routinely utilized for central and peripheral branch pulmonary artery stenosis (Figure 5), treatment of native and recurrent coarctation (especially older children and young adults) (Figure 6), systemic and pulmonary vein stenosis, right ventricular outflow tract intervention (as a standalone procedure to reduce obstruction or during conduit preparation for transcatheter pulmonary valve placement), as a framework for transcatheter valves, stenting of obstructed surgical baffles (following Mustard or Senning operation, Fontan pathways), creating and maintaining intracardiac shunts (atrial communications, Fontan fenestration creation, ductal stenting as source of pulmonary blood flow), as a hybrid strategy for single ventricle palliation to maintain systemic blood flow, and maintaining patency of surgical created shunts (stenotic systemic to pulmonary artery shunts). The basic principles of balloon expandable stent deployment are standard, yet there are numerous individual and institutional variations, and the interventional community has ‘MacGyvered’ techniques for the different situations. The patient is therapeutically anticoagulated. Depending on the perceived risk, cross-matched unit of blood is available in the catheterization laboratory. An end-hole catheter is advanced distal to the intended implantation site and exchanged over a guidewire for a long sheath placed distal to the implantation site. The stent is advanced via the long sheath and centered across the stenotic area and unsheathed. After confirming accurate positioning, the stent is then deployed by inflating the balloon. Hemodynamic and angiographic assessments are performed to assess success and rule out complications.

Most pre-mounted stents have the advantage of having a smaller crossing profile (size of required delivery sheath), but the major limitation is the inability for them to be expanded to adult size (without intentional stent fracture). Given the large variation in patient size, the operator will frequently choose to use hand-mounted stents to allow for flexibility of balloon diameter, stent length, and other stent characteristics to match patient anatomy. Important limitations to hand-mounted stents includes a larger crossing profile and the potential for the stent to become unmounted as it is advanced to the implantation site (not common with pre-mounted stents as factory mounted stents are more secure). Strategies used to maximize the stability of the mounted stent on the balloon include using contrast as an adhesive on the balloon prior to mounting the stent, using umbilical tape for crimping, intentionally inflating the balloon so that the balloon is no longer well wrapped prior to mounting the stent, minimally inflating the balloon after the stent is crimped, and front-loading the stent (using the stent and balloon as a dilator) and advancing the long sheath with the stent already loaded.

When hand-mounted unexpanded stents are taken to a large diameter (larger than 10 mm) the stent first inflates at either end—the so called “dog bone” formation. As the stent is completely inflated, the sharp ends of the stent are directed into the vessel wall and can increase the risk of perforation and aneurysm formation. Fortunately, using a specialized balloon called Balloon-in Balloon (NuMED Inc., Hopkinton, NY, USA) addresses this problem and has greatly simplified large diameter stent deployment. The inner balloon which is one half the diameter of the outer balloon and a centimeter shorter in length is inflated first. The inner balloon uniformly expands the stent (the operator can still reposition the stent if necessary). The outer balloon is then inflated to deploy the stent at the intended final diameter. Other technical improvements include newer kink-resistant long sheaths and stiff exchange length guidewires to support advancement of large diameter stents through tortuous pathways. Finally, newer stent designs allow for stents to conform better to patient anatomy along curves, the open cell design allow for maintaining patency of intentionally jailed vessels, and large diameter balloon-expandable covered stents increase safety during treatment of severely stenotic lesions (Figure 7) and/or calcified vessels (coarctation of the aorta, right ventricular outflow tract conduits) and the ability to cover native or iatrogenic aneurysms.

Ductal stenting has emerged as a non-inferior alternative to surgical Blalock-Thomas-Taussig (BTT) shunt for palliation of ductal-dependent pulmonary blood flow (DDPBF) [63,64,65]. Ductal stenting for DDPBF is a technically challenging procedure with a definite learning curve. Cross sectional imaging with CT angiography is critical to understanding the anatomy, determine optimal vascular access site (typical arterial—carotid vs. axillary vs. femoral) based on site of ductal origin from the aorta, and to help determine high risk anatomies that may preclude ductal stenting or make it extremely challenging [9,65]. As the techniques have matured, there is recognition that all ductal tissue needs to be covered by the stent which by necessity requires protrusion of the stent into the aorta and the pulmonary artery; intentional jailing of the branch pulmonary arteries is common and often necessary. Both surgical and transcatheter approaches are associated with similar growth in pulmonary arteries and time to definitive surgical repair or single ventricle palliation. There is general agreement that ductal stenting is associated with shorter length of stay, lower diuretic use, and lower hospital charges albeit with a higher rate of reinterventions [64,65,66]. Some studies have shown lower mortality with ductal stenting for DDPBF compared with surgical BTT shunt while others have shown no difference [63,64]. A randomized head-to-head comparison between surgical BTT shunt and ductal stenting for DDPBF is desirable. Knowledge gaps include optimal stent choice and ductal anatomies that should be considered as contraindications for ductal stenting. Finally, a stent and delivery system specifically designed for ductal stenting is highly desirable.

Complications and knowledge gaps:i.Acute risks include access site vascular injury, stent migration, implantation site vascular injury, need for emergency surgery, compression of adjacent vascular structures (coronary artery compression during right ventricular outflow stenting or the airway, left bronchial compression during left pulmonary artery stenting after Fontan palliation), and rarely death.ii.In the intermediate and long-term, vascular stenosis secondary to neointimal proliferation, acquired stenosis secondary to somatic growth, and unintentional stent fracture are relevant clinical problems. Unintentional stent fracture results from cyclical loading imposed by the dynamic cardiovascular milieu at the implantation site and can be clinically silent or can result in stent collapse and vascular stenosis.iii.Fortunately, most stents can be safely dilated beyond manufacturer-recommended maximum diameter to accommodate for somatic growth (note that self-expanding stents cannot be dilated beyond nominal diameter). To accomplish this, serial incremental dilation is necessary. All stents dilated beyond the nominal diameter shorten longitudinally, with the degree of shortening being stent-specific and widely variable.iv.Small children are frequently treated with pre-mounted small and medium diameter stents owing to the necessity for a smaller crossing profile and flexibility. Here the only option for increasing stent diameter is surgical intervention or intentional stent fracture followed by placement of a larger stent. While feasibility has been shown on the bench and in some clinical studies, the generalizability of this approach is awaiting real world clinical data.

To address the major limitation of SI, relative stenosis secondary to somatic growth, several novel stent designs are being studied. Biodegradable stents are in pre-clinical and early clinical research [67,68,69]. These stents utilize either biocorrodable metals such as magnesium/iron or polymer based bioabsorbable scaffolds such as poly l-lactic acid (PLLA), tyrosine polycarbonate, polycaprolactone, or poly salicylic acid [3]. Current limitations include small size of currently approved adult stents, radial strength, radiopacity, variable rate of absorption, biological (inflammatory) response, distal embolization during resorption, and stent thrombosis [68,69]. Ewert et al. have reported animal studies using a balloon-expandable “growth stent” [70]. Two equal longitudinal halves of laser-cut stainless steel stent fragments are assembled into a circular stent using bioabsorbable sutures (facing tongue and groove elements prevent sliding of the two halves). As the sutures are absorbed, the separated halves will allow for future placement of an appropriate sized stent. Zahn et al. reported pre-clinical animal data with the Renata-Minima balloon-expandable, pre-mounted, cobalt chromium stent delivered via a proprietary delivery catheter (6 Fr crossing profile) designed to maintain radial strength over a range of diameters (4 to 22 mm) with predictable longitudinal shortening [71]. Finally, the Bentley BeGrow stent system is a L605 cobalt chromium, pre-mounted, balloon-expandable stent with designed controlled break points [72]. Once the stent is expanded beyond 11.5 mm, intentional fracture will be induced at the engineered break points. It is compatible with a 4 French sheath and 0.014-inch guide wire.

### 2.3. Transcatheter Valve Therapies

Most patients with CHD requiring surgical repair via either right ventricular outflow tract reconstruction or placement of a right ventricle to pulmonary artery conduit or bioprosthetic pulmonary valve ultimately develop right ventricular outflow tract dysfunction in the form of pulmonary regurgitation or stenosis. These defects include tetralogy of Fallot, truncus arteriosus, pulmonary atresia with or without VSD, many forms of double outlet right ventricle, as well as patients treated with the Ross operation. The hemodynamic consequence of chronic pressure and/or volume overload is progressive right ventricular dilation and dysfunction. Traditionally, surgical conduit and/or pulmonary valve replacement (PVR) has been the standard therapy in this instance. The first transcatheter pulmonary valve replacement (TPVR) was performed by Bonhoeffer and colleagues in 2000 and since then TPVR is a viable alternative to surgical PVR [73]. The Melody valve (Medtronic, Minneapolis, MN, USA) was the first FDA approved TPV for use within dysfunctional right ventricle to pulmonary artery conduits between 16 and 22 mm in diameter [74]. Subsequently, larger diameter valves (i.e., Sapien XT and Sapien S3 [Edwards Lifesciences, Irvine, CA, USA]) have become available [74,75,76]. Both these balloon-expandable TPVs have been used (off-label) to treat dysfunctional bioprosthetic and native or surgically altered pulmonary outflow tracts. While revolutionary, these TPVs only addressed ~25% of the dysfunctional right ventricular outflow tracts [77]. The newer large diameter self-expanding TPVs [Alterra pre-stent and Sapien 3 (Edwards Lifesciences, Irvine, CA, USA) and Harmony (Medtronic, Minneapolis, MN, USA)] were specifically developed for large diameter dysfunctional outflow tracts; these valves are now FDA approved [77,78,79,80].

TPVR is indicated for hemodynamically significant dysfunctional right ventricular outflow tracts (stenosis and/or insufficiency). For example, even in an asymptomatic tetralogy of Fallot patient with ≥ moderate pulmonary regurgitation, pulmonary valve replacement is indicated if any two of the following are present: (1) severe right ventricular dilation (RV end-diastolic volume > 160 mL/m^2^ or RV end-systolic volume > 80 mL/m^2^ or RV end diastolic volume > 2 × LV end diastolic volume), (2) mild to moderate RV or LV systolic dysfunction, (3) RV systolic pressure > 2/3 systemic pressure (secondary to RVOT obstruction), (4) progressive reduction in objective exercise tolerance [81]. All TPVR procedures require large bore venous access (16–26 Fr) and the dimensions of the conduit, bioprosthetic valve, or dysfunctional outflow tract are critical to device choice. All aspects of TPVR, from indication to device choice, are based on pre-procedural cross-sectional imaging (cMRI and/or gated cardiac CT). Balloon sizing is necessary for balloon expandable TPVs (Medtronic Melody TPV and Edwards-Sapien XT and S3 TPVs). Many conduits need to be “prepared” to receive the TPV with pre-stenting. There is controversy as to whether pre-stenting is necessary prior to placing a Sapien valve; if pre-stenting is used, the procedure may be staged due to concerns about displacing the freshly placed pre-stent. A critical step prior to pre-stenting is testing for coronary compression with balloon inflation. Once the conduit, bioprosthetic valve, or surgically altered right ventricular outflow tract is larger than 22–24 mm, the larger Sapien family of transcatheter heart valves are utilized [75]. When the outflow tract is larger than 29 mm, the large self-expanding valves are used (Harmony, Alterra-Sapien 3). Rapid ventricular pacing can be used for precise deployment when there is a short landing zone. For the large diameter self-expanding valves, sizing of large native/surgically altered outflow tracts is determined by sophisticated software programs that utilize computer modeling. A high-quality gated cardiac CT dataset is used to determine device fit and feasibility of TPVR prior to the actual procedure.

Complications and Knowledge Gaps:i.Endocarditis: The risk of endocarditis is highest with the Melody valve and lowest with the Sapien valves [82]. To date endocarditis has not been reported in Harmony TPV and Alterra pre-stent implants [82,83,84,85]. All TPVs have risk of endocarditis and therefore life-long endocarditis prophylaxis is recommended. Longer term data will be necessary to define patient- and device-related risk factors.ii.Growth strategies: For successful TPVR, defined as low resting gradient, the landing zone in conduits and bioprosthetic valves may not be of an adequate diameter. The ability to dilate the conduit beyond the nominal implant size and fracturing of the bioprosthetic valve ring can allow “growth” in some patients. The minimum diameter necessary and safety of this approach needs to be studied.iii.Anticoagulation: There is increasing recognition that valve function and possibly endocarditis risk are at least partly determined by thrombosis of the valve leaflets/housing. Optimal strategies for each valve have not yet been defined. Lifelong aspirin is increasingly used; some patients require dual anti-platelet therapy and/or anticoagulation.iv.Stent fractures: Prior to routine pre-stenting, Melody valve frame fractures were common. Fortunately, the incidence decreases with pre-stenting and adequate preparation of the conduit. The structural frame of the Sapien valve is stronger and frame fractures are rare. It is too early to know the risk for the new self-expanding valves.v.Pulmonary regurgitation: While the short and intermediate term freedom from reintervention for PR is generally good, longer-term data is necessary. For the larger self-expanding valves, the additional risk of perivalvular leak will require investigation. Currently, the incidence of significant perivalvular leak is small.vi.Arrhythmias: There appears to be a higher incidence of ventricular arrhythmia after TPVR especially with the larger self-expanding valves. Fortunately, a majority of the reported arrhythmias are benign (PVCs, non-sustained VT), respond to medications, and resolve in a majority of patients after several weeks [80]. We will need longer term data to understand the mechanism, risks, and treatment paradigm.

In the United States, there are now 4 FDA approved TPV therapies, Melody TPV (Figure 8), Harmony TPV (Figure 9), Sapien TPV (XT and S3) (Figure 10), and Alterra Adaptive Pre-stent with Sapien S3 TPV (Figure 11). Outside the US, other valve technologies are undergoing evaluation and include the Venus P-valve (Venus Medtech, Hangzhou, China) and the Pulsta valve (TaeWoong Medical Co., Gyeonggi-do, South Korea) [86,87]. Several pre-clinical and clinical efforts are underway to develop a living pulmonary valve with growth potential utilizing tissue engineering [88,89].

### 2.4. Vascular Occlusion Devices

In children with CHD, abnormal vascular connections are seen as a part of their cardiac defect or develop in response to hypoxemia or elevated pressures during staged single ventricle palliation [90]. Congenital abnormalities include systemic and pulmonary arterio-venous malformations, coronary artery fistulas, and portosystemic shunts. Acquired vascular malformations are commonly seen after the Fontan operation and include aortopulmonary collaterals, which develop in response to hypoxemia, and decompressing venous channels that develop in response to elevated venous pressure or obstructed systemic venous drainage [91,92,93,94]. Hemodynamically significant aortopulmonary collaterals are detrimental as they volume load the single ventricle [95]. Decompressing veno-venous collaterals can serve as a source for systemic desaturation and systemic thromboembolism [94]. Consequently, the interventional cardiologist is called upon to occlude these vascular malformations. Criteria for closure of aortopulmonary or venous collaterals are not universal and there is significant controversy regarding which patients are suitable candidates [93,96,97,98]. In general, a patient with a failing Fontan circulation with a hemodynamically significant burden of aortopulmonary collaterals, elevated end-diastolic pressure, and systemic AV valve regurgitation is a good candidate for occlusion [93]. Similarly, closure of decompressing venous collaterals with symptomatic systemic desaturation in an otherwise optimized Fontan circulation and acceptable hemodynamics is also less controversial.

Once the decision is made to occlude these channels, there are now several options available in the interventional toolbox. Many of these tools are used off-label but are standard clinical care in most institutions. Institutional and individual choice is a significant determinant when choosing a closure device. The most frequently used devices to occlude these abnormal vascular connections are coils, vascular plugs, embolization particles, and beads. There are numerous options when using coils for vascular occlusion. Coils can be free-release or detachable. Additional material such as microfibers and hydrogel are incorporate to maximize occlusion [99].

Many coils can now be delivered using microcatheters which make them attractive for use in complex and tortuous vessels. The Amplatzer Vascular Plug II and other Amplatzer family occluders have been used (off-label) for occlusion with excellent results in all forms of congenital and acquired lesions (Figure 12). Microvascular plugs (MVPs) are increasingly used as an off-label alternative to the Amplatzer family of devices. Several size options are available and have excellent trackability and deliverability; the small diameter MVPs can be delivered through a microcatheter. The biggest advantage of all vascular plugs is controlled delivery and retrievability prior to release. Particles and beads are flow-directed occlusion tools and are used for embolization of aortopulmonary collaterals (these are not to be used for venous collaterals). Their major advantage is that the material disperses into the distal vasculature and consequently lead to a lower incidence of recanalization, an important limitation of the other options [95]. The method of delivery is very different and great care is necessary to avoid unintended embolization which can have devastating ischemic consequences.

Complications and Knowledge Gaps:i.Acquired arterial and venous collaterals develop in response to abnormal hemodynamics. The clinical challenge is determining which patients will benefit from interventional closure and which patients may have no benefit and in rare cases may be harmed [93,96].ii.Coils and vascular plugs cause artifact during subsequent clinically necessary cross-sectional imaging. Fortunately, most of the devices are MR conditional and when appropriate care is taken can be safely imaged.iii.Recanalization after occlusion of aortopulmonary collaterals especially with coils and plugs is common. Recanalized vessels pose difficulty for repeat occlusion especially when significant. The optimal method to prevent recanalization remains currently undetermined [95].

## 3. Review the Role of Advanced Imaging and Its Integration in the Current Management of CHD

### 3.1. Echocardiography

Echocardiography has long been the standard for initial diagnostic evaluation of CHD. TTE, TEE, and ICE are imaging modalities used to guide transcatheter interventions. Interventionists are increasingly comfortable utilizing and interpreting echocardiographic imaging and integrating this technology to reduce use of fluoroscopy and limit radiation exposure.

TTE is mostly used to guide interventions in premature and small infants. When TEE is contraindicated or limited by patient size, TTE used for epicardial imaging in the operating room has been very useful. TTE has also revolutionized transcatheter PDA closure in premature infants by eliminating the need for arterial access for assessment of residual shunting and arch obstruction (Figure 4) [55]. TTE is increasing being used in infants and older children for certain procedures such as PDA and ASD closure [100]. The major limitation of TTE is radiation exposure to the sonographer and limited access to acoustic windows under the sterile field. There is now widespread availability and experience using TEE for interventional guidance of transcatheter closure of ASDs and VSD, perivalvular leaks, and targeted transseptal puncture for left-sided interventions [101,102]. The need for continuous availability of an experienced echocardiographer during the procedure and lack of a dedicated 3D imaging probe for small children (<20 kg) are some logistical and technical limitations of TEE. While TEE can be performed with deep sedation in the older patient, careful attention is needed to protect the airway, and it can become quite uncomfortable for longer procedures. At least in the United States, most interventions are performed under general anesthesia, and therefore TEE continues to be commonly used. ICE allows for excellent imaging using a phased-array transducer mounted on the distal tip of a steerable catheter. These systems provide excellent near-field imaging of intracardiac structures. In the congenital interventional space, ICE has been utilized as an alternative to TEE for ASD/PFO closure, VSD closure, trans-septal puncture, and assessment after TPV placement (Figure 13) [103,104]. The major limitation to routine use of ICE in pediatric CHD includes higher cost, lack of a wider sector and 3D imaging, and need for additional 8–10 Fr venous access.

### 3.2. Three-Dimensional Imaging

For procedural guidance in CHD, where spatial relationships are key, 3D-imaging is vital. Three-dimensional echocardiography, cardiac CT, and cMRI all play a role in generating the necessary images and are dependent on available hardware and clinical expertise at a given site. Each modality has advantages and disadvantages, and these have been extensively reviewed [105,106]. Once a 3D dataset is obtained, it can be used to guide interventions in a variety of different ways. In this section, we will cover pre- and intra-procedural image guidance.

#### 3.2.1. Pre-Procedural Guidance

In pre-procedural guidance, the goal for the imaging and interventional team is to develop a shared mental model of the anatomy and develop a roadmap for the procedure. Currently, the standard approach is to review echo and cross-sectional imaging on a two-dimensional display (computer screen). With the advent of three-dimensional technologies, new methods are now in clinical use.

#### 3.2.2. Three-Dimensional Printing

3DPAM not only helps visualize anatomical and structural relationships, but also allows the implanter to “practice” the procedure on the model [10,107,108]. Generating a 3DPAM is a complex process and several overviews are available [109,110,111]. The key elements in obtaining a reliable and accurate model are obtaining excellent high resolution source images, accurate and timely image segmentation, and choosing the right printing material based on the clinical need. Examples where 3DPAM have been used include covered stent repair of SVASD, aortic arch interventions, TPVR, pulmonary venous baffle stenting after an atrial switch operation, coronary interventions, and closure of ruptured sinus of Valsalva aneurysm [10,112,113,114,115,116,117].

#### 3.2.3. Virtual Reality

For this paper, we define virtual reality as a technology that generates a 3D representation of a structure with depth perception and is accomplished with either a 3D flat display or a head-mounted display. Given that virtual reality is a newer technology, there is significantly less literature regarding its use in transcatheter/percutaneous intervention planning. Virtual reality lets the user be more immersed in the anatomy, allows multiple virtual “cuts”, and may save time compared to 3DPAMs, but currently lacks the tactile and haptic feedback a 3DPAM may provide. Virtual reality has shown utility in planning transcatheter sinus venosus defect closure, hybrid transcatheter/operative complex ventricular septal defect closure, and ventricular assist device sizing and placement (Figure 14) [118,119,120].

### 3.3. Intraoperative/Intraprocedural Guidance

Traditionally, real time angiography helps to create the relevant anatomy during an interventional catheterization procedure. The interventionist “remembers and creates” a mental roadmap based off the angiograms and manipulates catheters, wires, and devices using this abstract roadmap. As a guide, a stored image is shown next to the live image. The goal of imaging-based intraprocedural guidance is to show preoperative or real-time anatomy as an overlay during active manipulation of catheters, wires, and interventional devices on the fluoroscopy screen replacing the mental roadmap described above.

#### 3.3.1. Echocardiography-Fluoroscopy Fusion

A key recent development in intra-procedural guidance is echocardiography-fluoroscopy fusion (EFF). In this technique, the real-time echocardiographic image, generally transesophageal echocardiography (TEE), is overlaid onto the fluoroscopic image in the correct orientation. In EFF, the TEE probe is registered on the fluoroscopy screen, to allow real-time reorientation of TEE images. Markers are then placed in the 3D fluoroscopy space and tracked during the procedure. TEE images are automatically rendered in 3D and overlaid on the fluoroscopy screen. Thus, the interventionalist does not need to change spatial orientation in their mental model of the anatomy and can see the catheters, outlined by fluoroscopy, in the rendered TEE anatomy [121,122]. The technology has been used for several interventions including device closure (ASDs, PFOs, VSDs, Fontan fenestration) and TPVR [122,123].

##### CT/cMRI Overlay

Pre-procedural cross-sectional imaging is now routinely performed. The CT and cMRI 3D datasets can be repurposed and overlaid on the live fluoroscopy screen (Figure 15) [11,108,124,125,126]. A registration algorithm is used to ensure that the overlay is accurately illustrated on the live fluoroscopic screen. Some of the factors that can cause registration mismatch include respiratory variations, anatomic distortion related to changes induced by stiff interventional guidewires, and patient positional patterns (imaging is typically obtained with the arms on the side, whereas the arms are frequently positioned over the head to remove them from the lateral fluoroscopy field of view). Some of these registration challenges can be overcome by using a 3D dataset generated in the catheterization suite with rotational angiography.

#### 3.3.2. Augmented Reality Real-Time Guidance

Augmented reality is distinct from virtual reality where patient-specific data is displayed superimposed on the real-world visual field. This can be used to superimpose anatomy or guide instructions directly “on” the patient in the operative or interventional field, rather than the image fusion techniques that display images on the fluoroscopy screen. Other uses include displaying physiologic or other patient specific data on demand during the procedure.

Intraprocedural guidance with augmented reality is already in use for spine and orthopedic surgery [127]. While still in its infancy, a few case reports describe AR for transcatheter aortic valve replacement and pacemaker implantation in complex congenital heart disease [128,129].

#### 3.3.3. Future Directions

The field of image guidance in CHD interventions continues to mature. Technological advances in raw computing power, temporal and spatial resolution, and registration algorithms should allow for increasing use of the technology during routine clinical care.

## 4. Current Regulatory Framework within the United States for Device Approval

Background: The lack of pediatric specific devices has been a long-standing problem [14]. Commonly cited reasons include lack of profitability to industry, cost for device development, heterogenous and small patient population, complex government regulations, long timeframe from conception to market, and need for lengthy clinical trials [14,15,130]. Out of necessity, the interventional community has creatively adapted available devices (off-label use) to serve the needs of the CHD population. The American Academy of Pediatrics in 2017 issued a policy statement supporting off-label use when appropriate [131]. Simultaneously, patient advocacy by parents and families, medical societies, registries, and academia have been successful in lobbying for development of pediatric-specific products. Both regulatory agencies and industry have responded to these needs [13,15].

In the United States, the Food and Drug Administration (FDA) has made efforts to foster development of innovative pediatric-specific devices. Several programs aimed at improving efficiency and creating an environment conducive to the development of pediatric-specific devices have been implemented. The FDA Safety and Innovation Act, the 21st Century Cures Act, the Early Feasibility Study Program, and the Humanitarian Device Exemption Program are noteworthy in this area [13]. For example, to reduce the burden of clinical trials, device manufacturers can now utilize available clinical data generated during routine clinical care (well conducted case series, multi-institutional studies, and medical registry data), extrapolate data from adult studies and trials, and utilize virtual patient simulations or implants as supportive data during the regulatory approval process [13].

Internationally, there are efforts to align regulatory requirements among countries to maximize safety and simultaneously eliminate inefficient regulations, thereby shortening the regulatory approval process [13,132]. These guiding principles were used to successfully launch the Harmonization By Doing (2003) and Harmonization By Doing-for-Children (2016) programs as a partnership between stakeholder from academia, industry, and regulatory agencies in the United States and Japan [15]. Approval of the Harmony™ TPV serves as an excellent real-world example of how the stakeholders leveraged the current patient friendly regulatory environment. The Harmony™ Transcatheter Pulmonary Valve (TPV) system (Medtronic, Minneapolis, MN, USA) was first evaluated in the United States using the FDA Early Feasibility Study Pathway [85]. Then, Harmonization By Doing-for-Children program initiated a global trial to evaluate the safety and efficacy of the Harmony™ (TPV) System [15]. The Harmony™ TPV was approved in March 2021 (USFDA PMA approval) and launched for clinical use.

## 5. Conclusions

Advances in technology and innovative transcatheter techniques have brought about a significant change in the management protocols for CHD. Transcatheter implantation of devices and/or valves has become first line therapy for many CHDs hitherto treated surgically. While off-label use continues to be common practice, the number of dedicated purpose-built devices with a CHD indication continues to increase. The field has evolved from the MacGyver pediatric interventionist innovating and repurposing devices to treat patients, to a rigorous and scientific process intended to bring innovative CHD specific and approved devices to market. Patient advocacy and a culture of collaboration at the regulatory level has set the stage for explosive growth of new CHD specific devices.

## Figures and Tables

**Figure 1 children-09-00497-f001:**
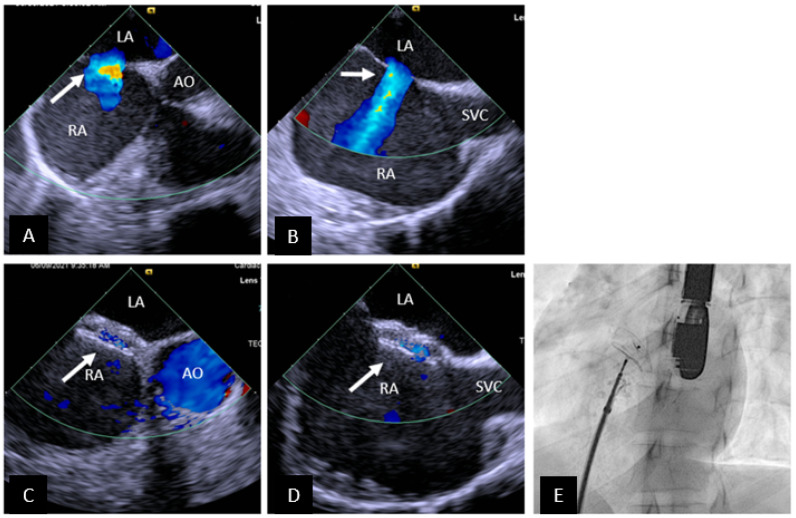
Transcatheter ASD closure with Amplatzer Septal Occluder. (**A**) 45° TEE view showing central secundum ASD (arrow) with good sized aortic rim. (**B**) 90° TEE view showing ASD (arrow) with good SVC rim. (**C**,**D**) Well positioned ASD device (arrow) in same views as (**A**,**B**) respectively. There is no residual shunting. (**E**) Fluoroscopic left anterior oblique view showing device position prior to release. SVC = superior vena cava, AO = aorta, RA = right atrium, LA = left atrium.

**Figure 2 children-09-00497-f002:**
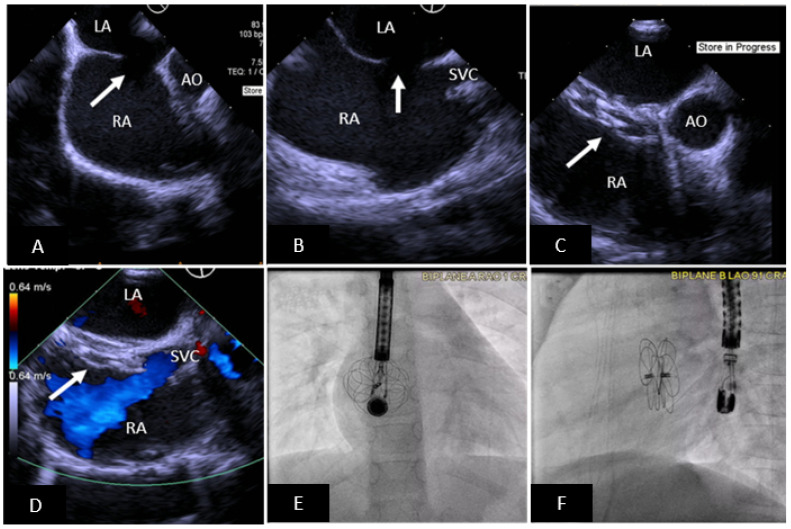
Transcatheter ASD closure with Gore Cardioform ASD Occluder. (**A**) 45° TEE view showing secundum ASD (arrow) with deficient aortic rim. (**B**) 90° TEE view showing high secundum ASD (arrow) with small SVC rim. (**C**) Well positioned ASD device (arrow) in same view as (**A**), the device does not indent the aortic root. (**D**) Well positioned ASD device in same view as (**B**) without obstruction to SVC flow. (**E**,**F**) Anteroposterior and lateral fluoroscopic images showing device position after release. SVC = superior vena cava, AO = aorta, RA = right atrium, LA = left atrium.

**Figure 3 children-09-00497-f003:**
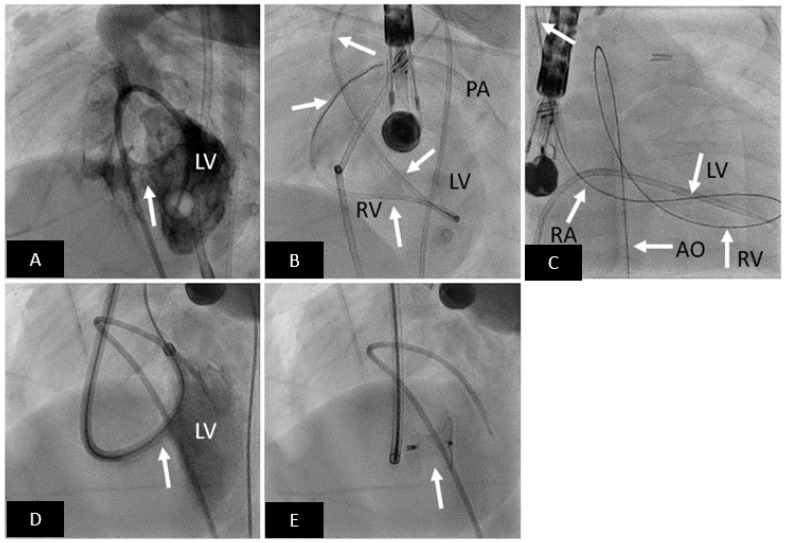
Transcatheter mVSD closure with Amplatzer Muscular VSD Occluder. (**A**) LV angiogram in cranially angulated left anterior oblique projection showing muscular VSD (arrow). (**B**) Catheter is advanced from aorta to LV, and then across the VSD into the RV, and then into the pulmonary artery. (**C**) An arteriovenous loop is created; right internal jugular-RA-RV-LV-AO-femoral artery. (**D**) Delivery sheath advanced from RIJ across the VSD (arrow) into the LV over the AV loop created. (**E**) Well positioned mVSD occluder (arrow) after release. LV = left ventricle, RV = right ventricle, PA = pulmonary artery, RA = right atrium, AO = aorta.

**Figure 4 children-09-00497-f004:**
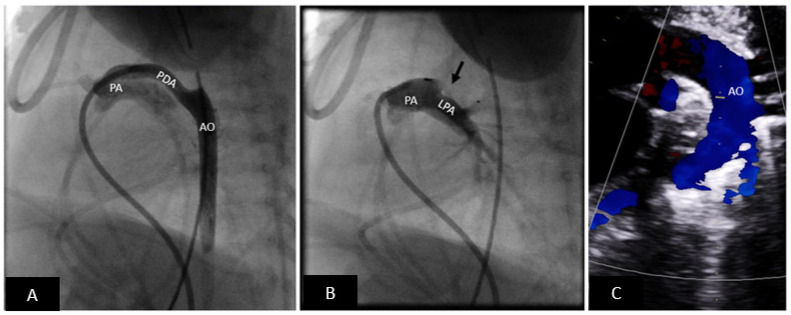
Premie PDA closure with microvascular plug (MVP). (**A**) Lateral angiogram showing a long tubular PDA. (**B**) PDA successfully closed with an MVP device (off-label use) (black arrow). The angiogram shows that there is no obstruction to the left pulmonary artery. (**C**) Transthoracic echocardiogram showing an unobstructed arch. There is no residual shunting. Echocardiographic assessment is critical to eliminate the need for arterial access. PA = pulmonary artery, LPA = left pulmonary artery, AO = aorta, PDA = patent ductus arteriosus.

**Figure 5 children-09-00497-f005:**
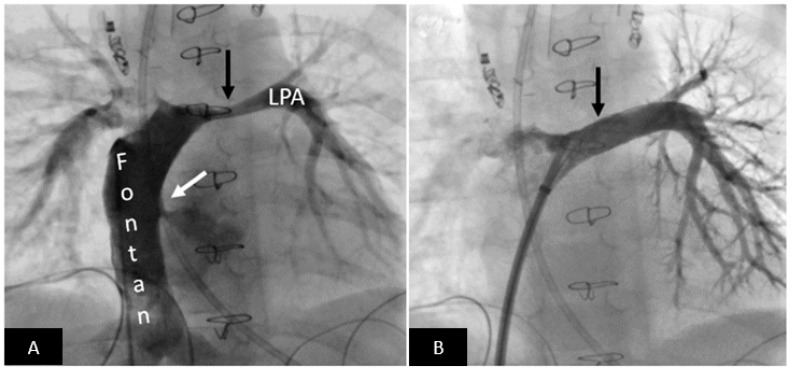
Postoperative left pulmonary artery stenting in a patient with persistent high-volume chest tube output after Fontan procedure. (**A**) Angiography in the Fontan pathway shows moderately severe long-segment left pulmonary artery stenosis (black arrow). Please note right to left shunt across the fenestration (white arrow). (**B**) Excellent angiographic result after stent implantation (arrow) resulting in resolution of the high-volume chest tube output. LPA = left pulmonary artery.

**Figure 6 children-09-00497-f006:**
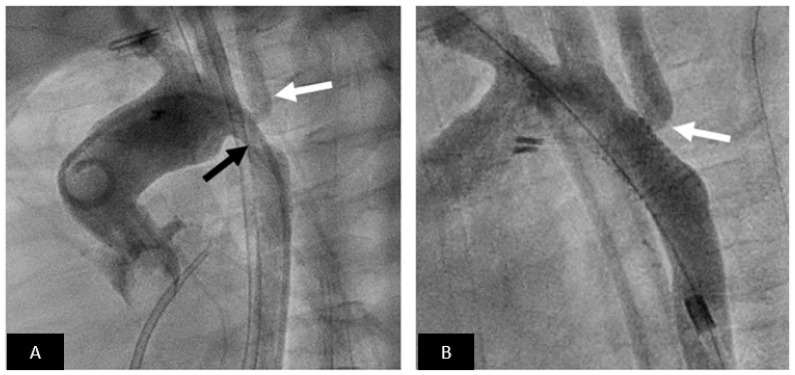
Coarctation stenting. (**A**) Aortogram in LAO view showing moderate coarctation (black arrow) involving the left subclavian artery (white arrow). (**B**) Excellent angiographic result following stent implantation without residual gradient. Note that even though the stent jails the left subclavian there is good flow (white arrow).

**Figure 7 children-09-00497-f007:**
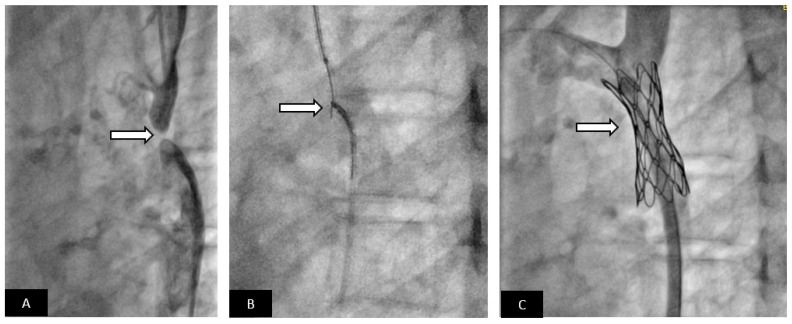
Covered stent for treatment of aortic coarctation with acquired atresia. Adolescent patient presented with severe systemic hypertension and exercise intolerance. (**A**) Lateral angiogram showing a short atretic segment (arrow). (**B**) A chronic total occlusion (CTO) guidewire is used to recanalize the atretic segment, a snare is placed in the distal aorta as a target for the CTO guidewire. Once the guidewire is snared, the segment is dilated with small angioplasty balloon. A long delivery sheath is placed. Using a pre-mounted covered stent increases the safety of the procedure compared to using a bare-metal stent. (**C**) Excellent angiographic result with a well-positioned covered stent without evidence of aortic wall injury; the patient had relief of systemic hypertension.

**Figure 8 children-09-00497-f008:**
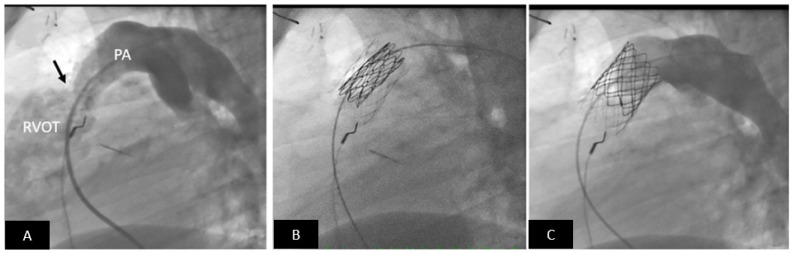
Transcatheter pulmonary valve replacement using a Medtronic Melody Valve. (**A**) Lateral angiogram showing a stenotic (arrow) RV-PA conduit; there is mild calcification. (**B**) The RV-PA conduit is prepared to receive the Melody valve by placement of a pre-stent (lighter stent frame). Also seen is the partially expanded Melody valve within the pre-stent (darker stent frame). (**C**) Excellent angiographic result with a well-positioned transcatheter Melody valve; note there is no pulmonary regurgitation. PA = pulmonary artery, RVOT = right ventricular outflow tract.

**Figure 9 children-09-00497-f009:**
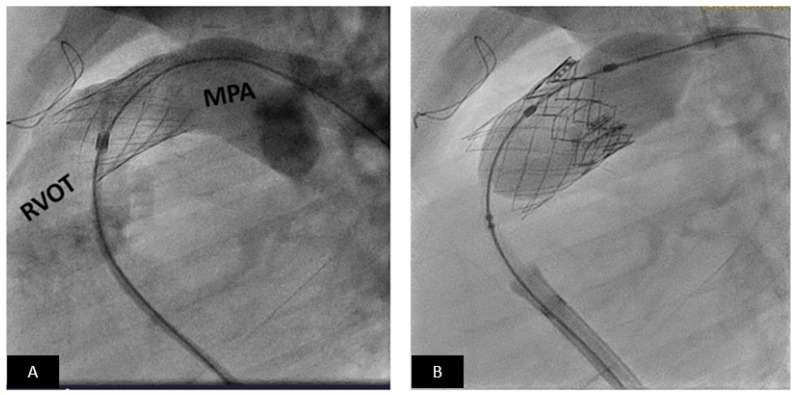
Transcatheter pulmonary valve replacement using an Edwards-Sapien XT valve. (**A**) Lateral angiogram showing a stented RVOT. (**B**) The angiogram shows Sapien valve (dark frame) deployment within the distal portion of the pre-stent. MPA = main pulmonary artery, RVOT = right ventricular outflow tract.

**Figure 10 children-09-00497-f010:**
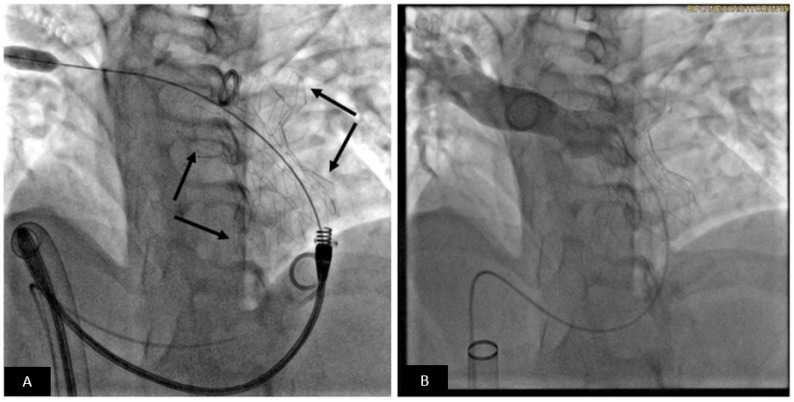
Transcatheter pulmonary valve replacement using Medtronic Harmony TPV. (**A**) Cranially angulated right anterior oblique projections showing the Harmony TPV (arrow) immediately after release. The delivery system is still across the valve. (**B**) Pulmonary artery angiogram shows unobstructed flow into the pulmonary artery branches. Note there is no pulmonary regurgitation.

**Figure 11 children-09-00497-f011:**
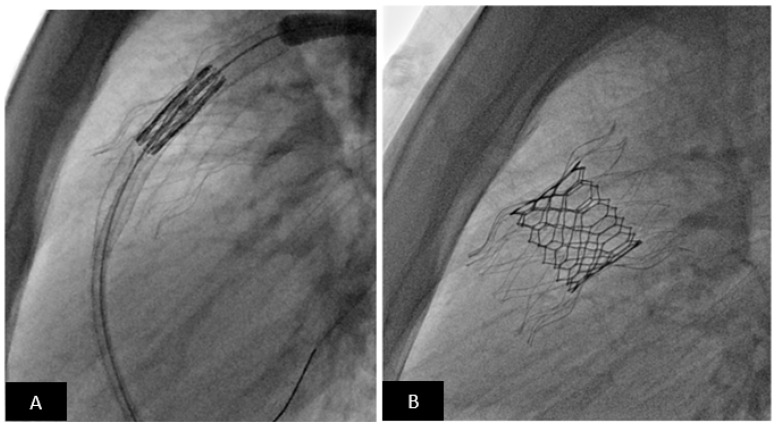
Transcatheter pulmonary valve replacement using Alterra pre-stent with Edwards-Sapien S3 TPV. (**A**) Lateral fluoroscopy projection showing Alterra Pre-stent (light stent frame) deployed. The Sapien S3 valve is positioned for deployment. (**B**) Lateral fluoroscopy projection showing the Sapien S3 valve (dark frame) deployed inside the Alterra Pre-stent (light frame).

**Figure 12 children-09-00497-f012:**
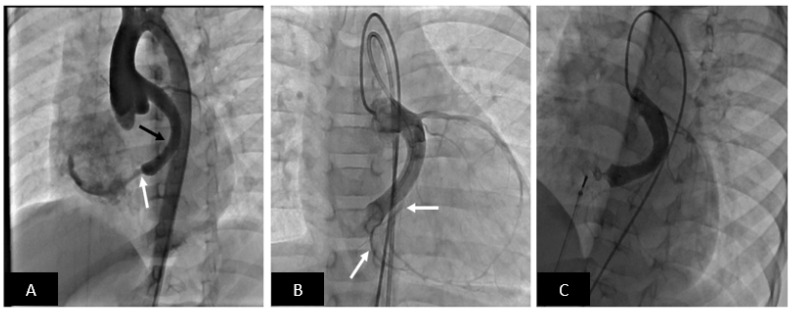
Transcatheter closure of large coronary artery fistula to the right atrium with Amplatzer Duct Occluder II (ADO-II). (**A**) Aortogram in cranially angulated left anterior oblique projection shows a coronary fistula from the left circumflex coronary artery into the right atrium (white arrow). The circumflex artery is dilated (black arrow). Note no coronary artery branches off the circumflex are visible. (**B**) Balloon occlusion coronary angiogram shows important coronary artery branches including a few just proximal to where the fistula enters the right atrium. This is of critical importance to avoid inadvertent occlusion of the branches by the occluding device. (**C**) The fistula is successfully occluded with an ADO-II (off-label use) device and no residual flow is seen. Note preserved flow into the terminal coronary artery branches.

**Figure 13 children-09-00497-f013:**
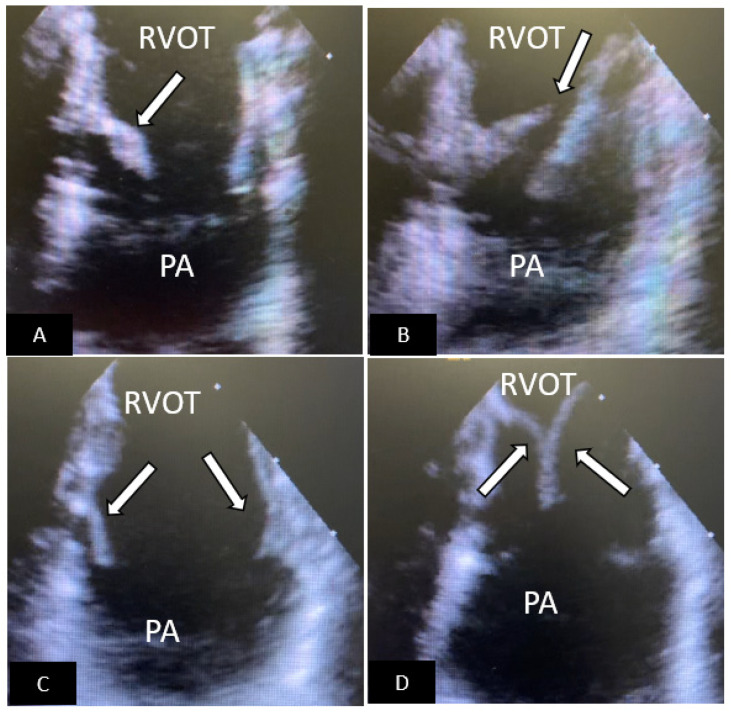
Intracardiac echocardiography (ICE) to assess for early failure of an Edwards-Sapien S3 TPV. The ICE probe is placed in the right ventricular outflow tract. (**A**) The TPV is seen in the open position. The medial leaflet (arrow) is shortened, thickened, and has limited systolic excursion. (**B**) When the TPV closes there is failure of coaptation of the leaflets (arrow) with resultant pulmonary regurgitation (not shown). (**C**) A new valve-in-valve Edwards-Sapien S3 was placed. ICE image now shows that the leaflets (arrows) are of the same length, thin, and show normal systolic excursion. (**D**) When the TPV closes the leaflets show excellent coaptation (arrow) and there is no pulmonary regurgitation (not shown). RVOT=right ventricular outflow tract, PA=pulmonary artery.

**Figure 14 children-09-00497-f014:**
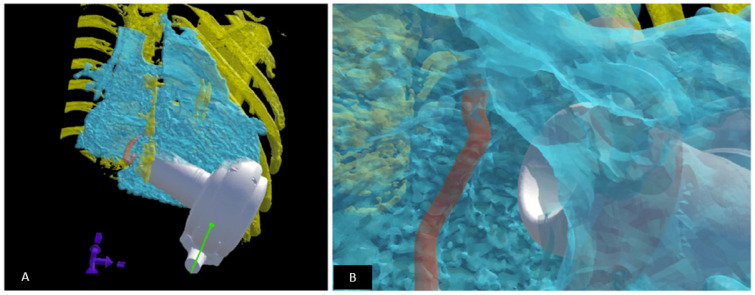
Fit testing for ventricular assist device (VAD) in children using virtual reality. VADs are not designed for children; those with complex heart disease have unique anatomical constraints which can limit a successful implant. A virtual implant can help determine the optimal VAD model and if an implant is feasible. (**A**) A virtual reality model is setup: the heart is created using blood pool from a CT dataset (depicted in blue). The rib cage (depicted in yellow) and the atrioventricular annulus (depicted in orange) are modelled to scale. The VAD (represented in white) is then virtually implanted as it would be during the surgical procedure within the rib cage. (**B**) The suitability of the fit is determined by “looking into” the ventricle and assessing the relationship to the AV annulus and the myocardium as well as the ability of the VAD to “sit” within the chest wall constraints. The implant is manipulated to determine optimal position during the virtual implant session. Revised from reference [118].

**Figure 15 children-09-00497-f015:**
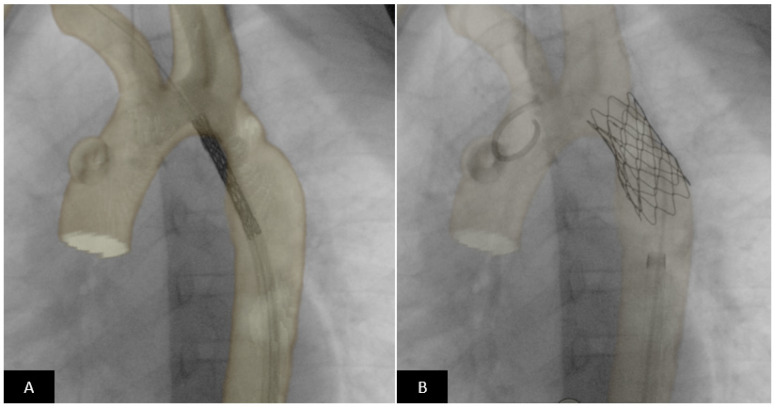
CT image overlay using VesselNavigator™ System (Philips Healthcare) for coarctation stenting. Pre-procedural CT dataset is overlaid on live fluoroscopy in the catheterization suite and used as a roadmap using image registration. (**A**) In real time the stent is positioned across the coarctation segment using the roadmap. Precise placement of the stent can be performed without repeated positioning angiograms based on the overlay image. (**B**) The stent is in excellent position completely covering the coarctation segment and positioned such that the origin of the left subclavian is not jailed.

## Data Availability

Not applicable.

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
