# Peer review of "Transcatheter Device Therapy and the Integration of Advanced Imaging in Congenital Heart Disease"

_children, 2022, doi:10.3390/children9040497_

Round 1

Reviewer 1 Report

This is an attractive topic to review hitherto transcatheter implantable devices for CHD, the role and integration of advanced imaging and explore the current regulatory framework for device approvals in USA. However, there are some points that should be discussed.

  1. Page 4, Line 128 mentioned removal of the device is recommended with any acute change in conduction, and the use of steroids for first degree heart block with occasional resolution of the block. However, in some studies, AV block improved spontaneously or resolved after steroids or device removal without recurrence at follow-up. Overall, the device removal or downsizing is recommended in the patients with high degree heart block after ASD device. Bink et al. reported an incidence of 15% of first-degree AV block in preoperative electrocardiograms of children with ASD. In the case of new first-degree AV block after cardiac catheterization, close observation is necessary so as not to overlook later progression to a higher degree of block. Authors should be clarified when steroid therapy is started (new first degree AV block? or high AV block?) and when the device needs to be removed (acute change in conduction? or persistent high degree AV block?).
  2. Page 5, Line 135 mentioned there are several other devices in Europe and Asia being used for septal defect occlusion. The Lifetech CeraFlex ASD occluder (LifeTech Scientific Co., Shenzhen, China) also received CE Mark Approval. A comparative study between Ceraflex and ASO showed comparable success rate, safety and efficacy. All devices were successfully delivered and deployed (100%) [Astarcioglu MA, et al. Ceraflex versus Amplatzer occluder for secundum atrial septal defect closure. Multicenter clinical experience. Herz 2015;40 Suppl 2:146-50.].
  3. This article is rich in content, complete in structure and clear in paragraphs. However, the length seems to be too long. This will be difficult for readers. I would like to suggest deleting and modifying part of the content and length.
  4. There are typos in part of the content (eg. Page 7, line 206, Figure means what; Page 8, line 240, Figure 3 should be Figure 4; Page 11, line 361 CDH should be CHD) and need to be corrected. Authors would need to check it again carefully.

Author Response

The authors would like to thank the reviewers for their constructive critique of our manuscript.

  1. We have addressed different treatment options and outlined a graded approach in the event heart block occurs after ASD device closure.
  2. We have mentioned that the Ceraflex device is one of the available devices for ASD closure outside the US. We have also included the suggested reference.
  3. We have modified both the length and content to improve readability of the manuscript.
  4. We have checked the manuscript for errors as suggested and revised the entire manuscript. We have checked that the figure numbers in the text refer to the correct accompanying figure.

Reviewer 2 Report

I have reviewed the manuscript entitled "Advances in transcatheter device technology for use in congenital heart defects" and, after extensive revision, consider it publishable in principle. It is instructive and beautifully illustrated for the most part. In my opinion, the readership could benefit from a revised version if more details are included, and the authors act with more care with regard to language and spelling. Not being a native speaker myself, I feel that there are clearly too many careless errors in figures and text.

The intent of the manuscript is to provide the reader with a comprehensive overview of the interventional devices and therapies currently available for the treatment of congenital heart defects. This is only partially achieved in the current version as important new approaches have not yet been considered. For example, ductal stenting as a relatively new option and alternative to surgical shunt surgery has not yet been mentioned. Also, the interventional treatment of sinus venosus defects should, in my opinion, at least marginally be worth a note. Finally, these are the most recent advances in the interventional treatment of children with CHD.

In terms of structure, the authors have combined three topics (devices, imaging, regulations), of which only one is reflected in the title. The title should be changed accordingly and at least include imaging.

I find the device and imaging part well summarized, whereby the imaging part should also be illustrated.

The regulation part is, in my opinion, much too long. It does not have to disappear completely but should be shortened to half a page.

I am willing to review the manuscript again after a major revision.

Author Response

The authors would like to thank the reviewers for their constructive critique of our manuscript.

  1. We have had the manuscript grammatically revised by a native English speaker. We have corrected the many mistakes pointed out by the reviewer.
  2. We have included a section on ductal stenting and transcatheter repair of SVASD repair. We think that this was an excellent suggestion and improves the manuscript.
  3. The title has been changed to include imaging.
  4. We have added additional images to illustrate the imaging section of the manuscript.
  5. We have condensed the regulatory review substantially. 

Round 2

Reviewer 2 Report

I congratulate the authors for the excellent work!
Thank you for incorporating my comments. I have a few formal comments.
Lines 68-74: I think you either end each bullet point with or without a period- but always the same.
Lines 84-87: The font size of the subheadings is smaller than above. I think you either capitalize or lowercase each word- but always the same. 
Line 109: The ® is sometimes in a circle, sometimes without, in other marks not at all. It should be the same throughout the manuscript. A comma is missing after Flagstaff.
Lines 116-119: Please make the sentence clearer.
Line 147: EKG is the German term for what you mean.

About the illustrations:
In general very nice and instructive!!!
The labels distributed over the figures are very inhomogeneous. Sometimes top left, sometimes bottom left. Sometimes white underlaid sometimes not should be the same everywhere! The sequence of images in Figure 2 is irritating and can be changed. 

In the headings and subheadings, sometimes every word is written in capital letters, sometimes in small letters- you know.....

Author Response

Response to comments from the reviewer:

Lines 68-74: I think you either end each bullet point with or without a period- but always the same. (Each bullet point now ends with a period.)
Lines 84-87: The font size of the subheadings is smaller than above. I think you either capitalize or lowercase each word- but always the same. (Uniform approach – sentence case is used.)
Line 109: The ® is sometimes in a circle, sometimes without, in other marks not at all. It should be the same throughout the manuscript. A comma is missing after Flagstaff. (Based on our review, if the journal does not require use of trademark symbols, they should be omitted. We have therefore decided to remove all trademark symbols. The trademark symbols have been retained in the references to make sure they match the title of the manuscript accurately. A comma added after Flagstaff.)
Lines 116-119: Please make the sentence clearer. (This sentence is indeed difficult to read; we have simplified it to state -- Both devices approved for ASD closure in the United States have demonstrated excellent safety and efficacy.)
Line 147: EKG is the German term for what you mean. (Replaced with electrocardiogram)

About the illustrations:
In general very nice and instructive!!!
The labels distributed over the figures are very inhomogeneous. Sometimes top left, sometimes bottom left. Sometimes white underlaid sometimes not should be the same everywhere! The sequence of images in Figure 2 is irritating and can be changed. (Sequence of Figure 2 has been changed and it is now easier to read. The labels as now all with a black background with white lettering; they are all at the bottom left corner of the images.)

In the headings and subheadings, sometimes every word is written in capital letters, sometimes in small letters- you know..... (Uniform structure – sentence case is used.)